# Pharmacological Evidence Suggests That Slo3 Channel Is the Principal K^+^ Channel in Boar Spermatozoa

**DOI:** 10.3390/ijms24097806

**Published:** 2023-04-25

**Authors:** Akila Cooray, Jeongsook Kim, Beno Ramesh Nirujan, Nishani Jayanika Jayathilake, Kyu Pil Lee

**Affiliations:** Department of Physiology, College of Veterinary Medicine, Chungnam National University, Daejeon 34134, Republic of Korea

**Keywords:** boar sperm, Slo3, ion homeostasis, LDD175, motility, acrosome reaction

## Abstract

Sperm ion channels are associated with the quality and type of flagellar movement, and their differential regulation is crucial for sperm function during specific phases. The principal potassium ion channel is responsible for the majority of K^+^ ion flux, resulting in membrane hyperpolarization, and is essential for sperm capacitation-related signaling pathways. The molecular identity of the principal K^+^ channel varies greatly between different species, and there is a lack of information about boar K^+^ channels. We aimed to determine the channel identity of boar sperm contributing to the primary K^+^ current using pharmacological dissection. A series of Slo1 and Slo3 channel modulators were used for treatment. Sperm motility and related kinematic parameters were monitored using a computer-assisted sperm analysis system under non-capacitated conditions. Time-lapse flow cytometry with fluorochromes was used to measure changes in different intracellular ionic concentrations, and conventional flow cytometry was used to determine the acrosome reaction. Membrane depolarization, reduction in acrosome reaction, and motility parameters were observed upon the inhibition of the Slo3 channel, suggesting that the *Slo3* gene encodes the main K^+^ channel in boar spermatozoa. The Slo3 channel was localized on the sperm flagellum, and the inhibition of Slo3 did not reduce sperm viability. These results may aid potential animal-model-based extrapolations and help to ameliorate motility and related parameters, leading to improved assisted reproductive methods in industrial livestock production.

## 1. Introduction

The successful fertilization of an egg by a sperm depends on a series of prior events that alter sperm morphology and physiology. These changes are initiated by different external stimuli in the male and female reproductive tracts, which, in turn, regulate the ion homeostasis of the sperm [1]. Ion channels and transporters are present in sperm cells to identify these external cues to maintain optimal ion concentrations relevant to events such as capacitation-associated hyperpolarization, internal alkalization, and acrosome reactions [2]. Because the speed of ion movement across ion channels is greater than across transporters, ion channels can respond to external cues more rapidly. Calcium, potassium, protons, and sodium are the main cations transported across the sperm membrane, while chloride and bicarbonate are the major anions [3]. Some of the most well-known major ion channels in sperm are the cation channel of sperm (CatSper) and the voltage-gated proton channel (Hv1), which are responsible for the uptake of calcium and proton extrusion, respectively.

Capacitation is a vital step in the journey of sperm until fertilization and is dependent on sperm membrane hyperpolarization. In capacitated sperm, the membrane potential is more hyperpolarized than in non-capacitated sperm, which is attributed to increased K^+^ permeability across the sperm membrane as membrane hyperpolarization occurs primarily due to the efflux of potassium ions via the principal potassium ion channel [4]. In addition, accumulating evidence in the literature suggests that the acrosome reaction in sperm also depends on membrane hyperpolarization [5,6]. The deletion or inhibition of the ion channels contributing to primary K^+^ flux in spermatozoa negatively affects their ability to complete fertilization [7,8,9].

To date, research studies have reported two different molecular identities of significant channels for K^+^ flux from the sperm: Slo1 and Slo3. Although Slo1 and Slo3 share high evolutionary homology, they show a marked difference in their functional characteristics and tissue distribution. Slo1 is expressed in many tissues, including muscle, neuronal, and reproductive tissues. Slo3 is mainly expressed in mammalian male reproductive tissues [10]; however, a recent study by Chávez et al. found cytoplasmic Slo3 expression in somatic tissues [11]. Considering the ion channel properties, Slo1 is activated by an intracellular increase in Ca^2+^ and Mg^2+^ levels and voltage [12]. Slo1 is independent of intracellular pH changes [13], whereas Slo3 channel is strongly dependent on intracellular alkalization and calcium levels [14,15,16]. However, the precise mechanism underlying the regulation of human Slo3 remains unclear. The literature supports the idea that Slo3 is gated by intracellular pH, intracellular calcium, or both [15,17,18,19,20].

Despite considerable research, there is controversy regarding the principal K^+^ channel of human spermatozoa. The identity of the main K^+^ channel in humans is suggested to be Slo1 [13]. Slo3 in human spermatozoa also plays a vital role in sperm physiology, as homozygous mutations in Slo3 result in severe asthenozoospermia and selective inhibition impairs sperm function [18,21,22]. In mice, electrophysiological data favor Slo3 as the principal route of K^+^ conductance [9,23]. Capacitation and acrosome reactions in swine sperm are also strictly dependent on potassium channels [24]. More comprehensive data need to be collected to identify the primary K^+^ channel contributing to the outward K^+^ channel in boar spermatozoa, and the literature suggests that Slo1 is important for capacitation, motility, and acrosome reactions [25].

One of our previous reports demonstrated that the differential modulation of the human Slo3 (hSlo3) channel by BK_ca_ activators allows for pharmacological discrimination between Slo1 and Slo3 channels. LDD175 (4-chloro-7-trifluoromethyl-10H-benzo[4,5]furo[3,2-b]indole-1-carboxylic acid; also known as CTBIC) enhances hSlo3, while the extracellular application of NS1619 (1,3-dihydro-1-[2-hydroxy-5-(trifluoromethyl)-phenyl]-5-(trifluoromethyl)-2H-benzimidazole-2-one) inhibits the Slo3 channel [26]. Furthermore, studies on the differential pharmacological properties of K^+^ channels have shown that iberiotoxin (IbTx) and paxilline (PAX) selectively inhibit Slo1 [27,28].

Despite the significance of K^+^ ion flux in sperm physiology and the differences observed between human and animal sperm K^+^ channels, the principal K^+^ channel of sperm in other species is relatively less explored. Against the background of the different pharmacological properties of the principal K^+^ channel candidates, this study was designed to provide insight into the principal K^+^ channel identity of boar spermatozoa, relating to their motility. For this purpose, we tested changes in intracellular parameters, including membrane potential, pH, and calcium, using K^+^ channel modulators. Our results suggest that Slo3 is predominant over Slo1 in boar spermatozoa hyperpolarization, and the inhibition of Slo3 reduces sperm motility and acrosome reactions. Furthermore, calcium entry from the extracellular space into sperm is Slo3-dependent, which suggests its control over CatSper, the main calcium entry into sperm.

To the best of our knowledge, this is the first study to report the identity and pharmacological characterization of the principal K^+^ channels in boar sperm. K^+^ channels in the sperm membrane are important for sperm physiology. Therefore, the identification and characterization of the principal K^+^ channel in different species have significant implications for the detailed understanding of sperm physiology and potential animal-model-based extrapolations, thereby increasing sperm quality and overcoming the deleterious outcomes of fertility-related pathologies.

## 2. Results

### 2.1. Effect of K^+^ Channel Modulators on Boar Sperm Motility and Kinematic Parameters

Progressive motility and the rapid motile sperm count showed a pattern similar to that of total motility, with LDD175 treatment causing an increase and NS1619 treatment causing a decline. The 5 min incubation of 10 µM of LDD175 produced the highest progressive motility and rapid sperm count (55% and 50%, respectively). In contrast, treatment with 100 µM of NS1619 for 5 min resulted in the lowest progressive motility and least rapid sperm at nearly 10% (Figure 1). The total motility of the sperm (total motility% = 100 − immotile%) increased with LDD175 treatment up to nearly 80% when incubated for 5 min. The increase in motility after the LDD175 treatment was concentration-dependent, with the maximum motility observed after treatment with 10 µM of LDD175. The motility of sperm treated with NS1619 for 5 min continuously decreased with increasing concentration, and the least motility was observed, approximately 45% after treatment with 100 µM of NS1619. 

### 2.2. LDD175 Treatment Hyperpolarizes and NS1619 Treatment Depolarizes the Sperm Membrane, While Other Slo1 Modulators Do Not Have a Significant Effect

Normalized 3,3′-dipropyl thiadicarbo cyanine iodide (DiSC_3_(5)) fluorescence upon treatment with LDD175 increased stepwise up to 10 µM, and the further addition of LDD175 resulted in a decrease in DiSC3(5) fluorescence, but the fluorescence at any tested concentration was not significantly lower than the basal fluorescence level. No significant changes in fluorescence corresponding to membrane potential were observed upon treatment with up to 10 µM of NS1619. A rapid and significant decrease in DiSC_3_(5) fluorescence was observed after treatment with 50 µM of NS1619, and the fluorescence further decreased after the cumulative addition of 100 µM (Figure 2A). Compared with the LDD175- and NS1619-treated groups, the vehicle control-treated group (DMSO) showed no significant changes in E_m._ DiSC_3_(5) fluorescence did not change with iberiotoxin (IbTX) and paxillin (PAX) treatments at concentrations of up to 200 nM (Appendix A). The fluorescence gradually increased with PAX, but the increase was not dose-dependent. The sequential addition of PAX (100 nM) (Figure 2E,F) or IbTx (100 nM) (Figure 2C,D) before and after LDD175 treatment resulted in a slight decrease in DiSC_3_(5) fluorescence. The changes in fluorescence after the PAX and IbTx treatments were statistically insignificant compared with the fluorescence of the same settings immediately before treatment. In both scenarios, the LDD175 treatment significantly increased DiSC_3_(5) fluorescence, regardless of the effects of IbTX and PAX (Figure 2C–F).

### 2.3. Intracellular Calcium Signals Change with K^+^ Channel Modulations

Normalized Fluo4-AM fluorescence in a Non-CAP medium containing 1.8 mM of Ca^2+^ increased with the extracellular application of LDD175 (Figure 3A). This increase was evident even after treatment with 1 µM of LDD175, and the highest increase was recorded when the cumulative concentration of LDD175 was 5 µM. This was followed by a gradual decrease in the basal fluorescence levels upon further LDD175 addition. Treatment with 50 µM of NS1619 in the Non-CAP medium also resulted in a sudden increase in intracellular calcium fluorescence, followed by a decrease with further NS1619 addition (Figure 3B). In the Ca^2+^-free medium, the intracellular calcium signals did not change after treatment with up to 5 µM of LDD175. However, in contrast to the signals in the Non-CAP medium with external Ca^2+^, the fluorescence signals in the Ca^2+^-free medium were reduced when treated with 5–10 µM of LDD175, resulting in the lowest signal of nearly −1, corresponding to 100 µM of LDD175. The Fluo4-AM fluorescence signals patterns after treatment with NS1619 were also similar to those after treatment with LDD175 in the Ca^2+^-free medium, wherein concentrations above 50 µM resulted in reduced fluorescence. The normalized drop in fluorescence in the Ca^2+^-free medium was the highest in the LDD175-treated (10 µM) group, and it was twice as low as the lowest fluorescence corresponding to the NS1619 (100 µM) treatment in the Ca^2+^-free medium (Figure 3C).

In the 1 µM of NNC 55-0396-pre-treated groups, fluorescence gradually decreased with increasing concentrations (Figure 4). The decrease in fluorescence was obvious after treatment with 5 µM of LDD175 (A) and 50 µM of NS1619 (B). Similar to the NNC 55-0396-pre-treated groups, the NS1619 (D) group pre-treated with 60 mM of TEA also showed decreasing calcium signals with increasing concentrations. Treatment with 100 µM of NS1619 significantly reduced Fluo4-AM fluorescence, while treatment with LDD175 concentrations after TEA (C) did not significantly change intracellular calcium signal levels (Figure 4).

### 2.4. The Principal K^+^ Channel Affects Sperm Intracellular pH

The treatment with an increasing concentration of LDD175 caused a stepwise decrease in the pH-sensitive fluorescence, and a sudden, remarkable acidification was observed after treatment with 50 µM of LDD175; the lowest pH was observed after treatment with 100 µM of LDD175. The pH beyond the 5 µM LDD175 treatment was significantly lower than the resting pH of the spermatozoa. The effective concentration of LDD175 (10 µM of LDD175) only generated an approximately −0.2 drop in the pH compared with the basal level. Treatment with low concentrations of NS1619 (up to 10 µM) did not cause any significant difference in sperm [pH]_i_. However, pH was reduced by nearly 0.6 units after treatment with the final concentration of 50 µM of NS1619, and it further reduced to a relative unit of −0.8 after treatment with an accumulative concentration of 100 µM (Figure 5A,B). Neither IbTx nor PAX showed any significant differences in BCECF-AM fluorescence, indicating changes in pH within the tested concentration range (Figure 5C).

### 2.5. Inhibition of the Principal K^+^ Channel Reduces Acrosome Reaction

Negative control samples with no treatment resulted in nearly 17% acrosome-reacted sperm. In the LDD175 treated groups, the highest concentration resulted in the highest fluorescence, indicating a stronger acrosome reaction. Although the fluorescence after treatment with 10 µM of LDD175 was higher than that of its negative control counterparts, it was not significantly different. The NS1619 treatment showed a decreasing trend in the acrosome reaction after 100 µM, resulting in a nearly 14% decrease, which was significantly lower than the count obtained in the LDD175 (10 µM)-treated group (Figure 6). The acrosome-reacted sperm counts of all treatment groups were lower than that of the positive control (Appendix A).

### 2.6. Slo1 and Slo3 Channels Showed Different Localizations in Boar Spermatozoa

Immunofluorescence studies showed that most Slo1 channels were present in the head area with two concentrated regions, the anterior head region and the post-acrosomal region (Figure 7A). Slo3 channel fluorescence signals were less strong than Slo1 under the same confocal microscopic settings. However, a distinct area with apparent green fluorescence depicting the Slo3 channel was seen in the proximal tail area, and such fluorescence was absent in the head area (Figure 7B). Non-specific binding of the secondary antibody was not detected (Figure 7C).

### 2.7. Slo3 Channel Inhibition Improves Boar Cell Viability

After 5 min of incubation with LDD175 (10 µM) and NS1619 (100 µM), the intact live sperm population of both treatment groups was not significantly different from that of the negative control, and the live cell populations of NC, LDD175, and NS1619 were 52.9%, 49.5%, and 51.7%, respectively (Figure 7D). After 30 min of incubation, the values of NC and LDD175 slightly decreased, while NS1619 remained unchanged compared with its 5-min-incubated counterparts. Of the 30-min-incubated samples, the intact live cell population in 100 µM of NS1619 was significantly higher than in its negative control. After one hour of incubation with the treatments, LDD175 resulted in the lowest intact live cells, while the highest result was with NS1619, significantly different from the negative control (n = 5, *p* < 0.05).

## 3. Discussion

In recent decades, artificial insemination in pig farms has been preferred because of its efficacy, sustainability, lower chance of transmitting infections than in natural mating, and availability for genetic selection [29]. In addition, boar sperm are widely used as models in toxicological assays and for elucidating sperm physiology [30,31]. Therefore, detailed studies regarding boar sperm physiology are of great importance in improving the outcomes of these applications and increasing the accuracy of extrapolations. Membrane hyperpolarization is an important event in sperm capacitation in many species and may play a central role in mediating other events and cellular signaling pathways [32]. To date, studies have demonstrated that potassium channels are the critical ion channels responsible for this phenomenon. However, the molecular identity of the principal K^+^ channel in different species seems to be different, and despite the significance of the principal K^+^ channel in sperm physiology, the available data are elusive and limited to very few species. Two ion channels, Slo1 and Slo3, were identified as putative primary K^+^ ion-conducting channels in spermatozoa. Although ion channels such as voltage-gated potassium channels (K_v_) may contribute K^+^ channels and thereby maintain optimal spermatozoa physiology, evidence to date largely emphasizes K_v_s’ effects, mainly on sperm volume and related parameters [33,34,35,36,37]. Considering Slo channels, Slo3 facilitates major K^+^ conductance across the mouse sperm membrane [9], while both Slo1 and Slo3 play a significant role in the optimal conditioning of human sperm [17,38]. Yeste et al. demonstrated the significance of Slo1 in the capacitation and acrosome reactions of boar spermatozoa [24,25]. One of our previous reports showed that quercetin can inhibit human Slo3 current [19], and later, we found quercetin also depolarizes cryopreserved boar sperm [39]. This suggested that Slo3 in boar sperm may be functionally significant in maintaining its membrane potential. Findings to date suggest that the principal K^+^ (KSper) current is calcium-dependent and also that intracellular calcium levels can be regulated by the KSper channel [13,20,23]. Moreover, the intracellular pH of sperm can also modulate the KSper channel [20]. Taken together, these properties of the KSper channel in sperm agree with the properties of the Slo channels [17,20,27,40]. Nevertheless, these findings are insufficient to conclusively identify the principal K^+^ channel in pig spermatozoa. Hence, this investigation focused on pharmacologically differentiating the principal K^+^ channel identity in boar sperm. Based on the evidence available from the literature across different species and boar sperm regarding sperm membrane potential regulation, this study mainly focused on two channel identities, namely, Slo1 and Slo3. Using the pharmacological differences in Slo channels as evidence from previous studies, we used a series of K^+^ channel modulators to assess the identity of the principal K^+^ channel and its contribution to spermatic events. We used the dual actions of LDD175 and NS1619 to dissect the primary K^+^ channel identity of boar spermatozoa. Furthermore, we incorporated IbTx and PAX, known Slo1 inhibitors, in this study. 

Initially, we measured the changes in membrane potential after treatment with different K^+^ channel modulators. DiSC_3_(5), a membrane-potential-sensitive dye, was used to assess the intracellular K^+^ concentration. The rapid response (less than approximately 2 ms) to membrane potential makes DiSC_3_(5) suitable for identifying potential changes in time-lapse flow cytometry [41]. An increase in DiSC_3_(5) fluorescence in cells indicates membrane hyperpolarization. In agreement with previous studies on human [4] and mouse [42] sperm, only a subpopulation of boar sperm responds to membrane potential changes. Hence, the subpopulation responding to membrane potential was initially identified and then gated using a Valinomycin-KCl setup. Due to the limited sample size and variations between trials, we utilized relative changes in fluorescence in time-lapse flow cytometry for comparison purposes. The LDD175 treatment increased the fluorescence of spermatozoa up to a concentration of 10 µM, and maximum depolarization after treatment with NS1619 was recorded at a concentration of 100 µM. These data coincide with the concentrations corresponding to maximum Slo3 activation upon LDD175 treatment and inhibition upon NS1619 treatment in the patch-clamping recording, which were 10 µM and 100 µM, respectively [26]. These data suggest that Slo3 is important for maintaining membrane potential in non-capacitated boar sperm in vitro.

To further confirm the role of Slo3 in maintaining boar sperm membrane potential, we also assessed membrane potential changes with Slo1-specific blockers such as IbTx and PAX. According to the literature, Slo1-inhibiting concentrations of both chemicals are approximately 100 nM [13,23]. The membrane potential in groups treated with up to 200 nM of both chemicals remained relatively unchanged. These data, together with LDD175 and NS1619, suggest that the major K^+^ conductance of non-capacitated boar spermatozoa, resulting in membrane hyperpolarization, is mediated through Slo3 rather than Slo1. To identify potential residual traces of Slo1, membrane potential changes were measured after the sequential addition of LDD175 and Slo1 blockers. Despite the statistically insignificant changes, adding 100 nM PAX and IbTx before and after 10 µM of LDD175 resulted in a slight decrease in membrane potential, implicating potential minor K^+^ conductance through the Slo1 channels. In both scenarios with Slo1 blockers, 10 µM of LDD175 (Slo3 activation) generated significant membrane hyperpolarization, supporting the earlier claim that Slo3 could be the primary channel responsible for controlling the membrane potential of boar sperm. Derived from the data on membrane potential changes, we further examined the changes in motility and other major intracellular ion signals of boar spermatozoa with K^+^ channel modulators. 

Data from CASA showed a concentration-dependent improvement in motility-related parameters with LDD175, while a decreasing trend was observed with NS1619. Maximum and progressive motility were observed after treatment with 10 µM of LDD175. In contrast, the lowest motility was observed after the 100 µM NS1619 extracellular treatment. Numerous studies focusing on sperm of different species have concluded that the opening of the principal K^+^ channel is crucial for sperm hyperactivated motility, which is a result of capacitation [25,43,44]. Considering the concentrations corresponding to Slo3 activation by LDD175 and inhibition by NS1619 [26], this suggests that the activation of the Slo3 channel of boar spermatozoa increases its motility parameters. 

Intracellular calcium plays a central role in sperm physiology. Data from several studies unequivocally demonstrated that the elevation of intracellular calcium during the capacitation of boar sperm increases its motility parameters and tyrosine phosphorylation, which is the hallmark of sperm capacitation [45,46]. Membrane potential changes affect voltage-gated calcium ion channels in the cell, thereby altering the intracellular calcium levels. Several voltage-gated calcium-permeating channels, including CatSper, have been identified in mammalian sperm transmembrane [47]. Vicente-Carrilo et al. later reported the presence of a functional CatSper channel that is significant in boar sperm motility [48]. Against this background, we tested whether there were changes in intracellular calcium signals with the extracellular application of Slo1/3 channel modulators. 

Both LDD175 and NS1619 treatments in a Non-CAP medium increased intracellular calcium signals, and the increase with NS1619 was twice as high as that with LDD175. To determine whether the calcium increase was due to permeability changes in the transmembrane or calcium release from sperm internal sources, we repeated the same experiment but in an extracellular calcium-free medium. When extracellular calcium was removed, the increased calcium signals with both LDD175 and NS1619 in the Non-CAP medium were not detected. This demonstrates that the increase in intracellular calcium upon the addition of Slo1/3 modulators was extracellular-calcium-dependent, presumably due to the opening of calcium-permeating channels. Moreover, these results are in line with previous findings indicating that the increased motility and capacitation of boar spermatozoa were dependent on extracellular calcium and coincided with an increase in intracellular calcium signals [49,50]. 

Interestingly, although intracellular calcium levels were remarkably increased with higher concentrations of the NS1619 treatment, motility parameters showed opposite, decreasing trends. This could be explained by the dual action of calcium reported earlier, wherein low increases in calcium increased motility, whereas an increase caused by a relatively high amount (millimolar-scale) inhibited sperm motility [46,51]. Furthermore, as boar sperm express functional CatSper channels, which can be activated upon membrane depolarization, it is likely that the inhibition of boar Slo3 by NS1619 will result in sperm membrane depolarization, facilitating favorable conditions for the opening of boar CatSper. This increases intracellular calcium, corresponding to increased Fluo4-AM fluorescence signals. We tested the involvement of boar CatSper channels using membrane potential modulators. We postulated that if the membrane potential of the sperm affects calcium permeability through the CatSper channel, the calcium elevation of the CatSper-blocker-pre-treated sperm upon LDD175 or NS1619 treatment should be less than that without the CatSper blocker; therefore, CatSper-inhibitor-treated (NNC 55-0396) sperm were treated with LDD175 and NS1619 separately. When determining the concentration of NNC 55-0396, we considered the available electrophysiological data from the literature. A range of experiments shows that 1–20 µM of NNC 55-0396 can effectively inhibit CatSper currents in spermatozoa [52,53]. However, NNC55-0396 is a weak base, and extracellular application may alkalize sperm and thereby release calcium from its internal stores, resulting in an elevation in intracellular free calcium levels [54]. In agreement with this, in our experience (data not shown), calcium signals were elevated after treatment with NNC 55-0396 in a concentration-dependent manner. The elevation of calcium signals at 1 µM was insignificant compared with that in the untreated conditions. Hence, after the first two minutes of basal fluorescence recording in the time-lapse flow cytometry (TLFC), sperm were treated with 1 µM of NNC55-0396, followed by the sequential addition of either LDD175 or NS1619. The increase in fluorescence with low concentrations of LDD175 and NNC55-0396 was not as high as that in the non-treated group. Furthermore, NS1619, together with a CatSper blocker, showed no increase in intracellular calcium signals. These observations are consistent with our initial hypothesis that boar Slo3 modulates calcium’s entry into the cell, which mainly occurs through CatSper. Additionally, to rule out any possible direct effects of Slo1/3 modulators on the CatSper channel, the fluorescence of TEA-pre-treated sperm was measured. TEA at 60 mM inhibits both Slo1 and Slo3 [4]. In our experimental setting, neither the LDD175 treatment nor NS1619 treatment generated any significant elevation in [Ca^2+^]_i_ in TEA-pre-treated sperm. These results suggested that boar Slo3 is essential for potentiating CatSper, thereby regulating intracellular calcium levels. These findings are consistent with mouse sperm data showing that Slo3 channels can activate mouse CatSper through a voltage-dependent mechanism [32].

Acid extrusion is essential for sperm alkalization during capacitation, and Hv1 channels play a significant role in it [55]. The electrophysiological properties of the channel include activation via membrane depolarization. BCECF-AM, an intracellular pH-sensitive dye, was used to identify the changes in pH upon boar Slo channel modulation. When IbTx and PAX were added at a concentration of up to 200 nM, the pH of sperm remained unchanged, demonstrating that the Slo1 of boar sperm is not relevant for pH maintenance. With increasing LDD175 concentrations, the pH of the sperm became acidic, and the trend showed a typical stepwise dose response. Acidification upon LDD175 treatment could be due to the inhibition of Hv1 caused by membrane hyperpolarization; as boar sperm is found to have functional Hv1 channels that aid in proton (H^+^) efflux from sperm, sperm membrane depolarization is needed for the active functioning of the Hv1 channel; furthermore, in our initial experiments, we found that LDD175 can hyperpolarize the sperm plasma membrane. On the other hand, it was expected that NS1619, which inhibits Slo3, will make the sperm membrane depolarized, which, in turn, will activate Hv1 and alkalize the sperm. However, the fluorescence signals decreased even with the NS1619 treatment, indicating sperm acidification. The pattern showed no obvious dose–response. Regarding the relative drop in intracellular pH at the effective concentrations of the chemicals (NS1619 100 µM versus LDD175 10 µM), the highest drop corresponded to NS1619, which was four times lower than that of LDD175. It is noteworthy that, in addition to the Hv1 channel, Na^+^/H^+^ and Cl^-^/HCO_3_^-^ exchangers may contribute to maintaining sperm pH. In mouse sperm, membrane hyperpolarization itself increases the H^+^ outward channel via NHE. The exact role of these exchangers in boar sperm has not been fully explored. This may confound our understanding of boar-Slo3-dependent and pH-regulating sperm membrane components. In addition to the complexity of intracellular crosstalk, regional differences in pH were observed, where the acrosome was more acidic than the cytosol [56]. The elevation of cytosolic pH and calcium concentration and the release of acrosome calcium are known triggers of acrosome exocytosis. Considering these, together with the sharp increase in [Ca^2+^]_i_ upon NS1619 addition, the question of whether the acidification of sperm occurs in an acrosome-dependent manner remains unanswered. To expand our understanding and test the above assumptions, future research should focus on the effects of these chemicals on the established AR signaling pathways [57].

Capacitation followed by the acrosome reaction is essential for sperm physiology. Nevertheless, precision regarding the temporal and spatial occurrence of AR in the female reproductive tract is essential. Data from mouse studies suggest that membrane hyperpolarization itself is sufficient to complete the acrosome reaction [43]. In our study, we estimated the spontaneous acrosome reaction in the capacitating medium. The findings of our study are similar to those of mouse sperm studies, indicating that Slo3 increases the number of acrosome-reacted sperm. Although previous studies suggest that Slo1 is important in the boar sperm acrosome reaction [24,25], we did not identify any Slo1-specific changes in our settings. However, it is noteworthy that the difference in the studies is that previous experiments measured progesterone-induced acrosome exocytosis (10 µg/mL), whereas such an acrosome reaction induction was not employed in our study. Given that membrane hyperpolarization is itself sufficient [43,58] and a key regulatory factor for the completion of the acrosome reaction, we deduce that the trends observed regarding the acrosome reaction in the study would be the same but at a higher magnitude under acrosome-exocytosis-inducing conditions.

Through immunolocalization, the present study shows that the Slo3 channel of boar sperm is mainly located in the flagellum. The sperm tail is significant in its motility control, and the majority of motility-related ion channels are generally found in the principal piece of the sperm flagellum [1,48,59]. Additionally, most of these motility-associated channels are directly controlled by changes in K^+^ conductance through the sperm transmembrane. The physical proximity of the Slo3 channel to the sperm principal piece corroborates its functional relevance to sperm motility and associated signaling pathways. Yeste et. al. showed the presence of the Slo1 channel throughout the flagellum and anterior area of the post-acrosomal region of boar sperm [25]. However, we found Slo1 to be only localized in the head area. Furthermore, we also found Slo1 in the sperm apex area closer to the acrosome. This could be explained by the significance of Slo1 in the acrosome reaction, as evident in the literature [25].

To confirm if the decreased motility coincided with decreased sperm viability, we assessed sperm viability using an apoptosis assay. We used the maximally effective concentration of LDD175 and NS1619, and treated sperm were incubated for 5, 30, and 60 min. For the 5 min incubation, the intact live cell population was not significantly different from that of the negative control. However, as the incubation period increased, we observed that the inhibition of Slo3 can improve sperm membrane intactness, while Slo3-activated sperm showed the smallest population of intact live cells. This observation is consistent with the findings of Jean et al., who reported that the inhibition of the Slo3 channel in mouse and bull sperm can extend their viability [60]. Thus, it is safe to hypothesize that the motility reduction in sperm is not due to compromising its viability.

Based on these results, future studies using patch clamping to identify endogenous K^+^ currents will further validate the significance of Slo3 expression in boar spermatozoa. We also suggest considering other possible K^+^ currents, such as Na^+^/K^+^-ATPase, for fine-tuning the signaling cascade [61]. Furthermore, the current study utilizes pharmacological data, which may have been convoluted by the cross-reactivity of chemicals with different ion channels. Future patch-clamping experiments using chemicals with ion channels of interest will increase the reliability of these data.

Taken together, our results demonstrate that the role of boar Slo3 in sperm physiology is indispensable and that its effects are multi-fold, including changes in sperm *E_m_*, motility, [Ca^2+^]_i_, pH, and acrosome reactions. This study provides a significant contribution to the literature as the identification and mechanistic underpinning of the boar sperm principal K^+^ channel are paramount in improving the success of fertilization and sperm handling at animal farms, as evident from other animal studies (e.g., bull) [60], wherein specific blockers targeting Slo3 could extend fertilizing competence.

## 4. Materials and Methods

### 4.1. Reagents

Unless otherwise stated, all reagents and chemicals were purchased from Sigma-Aldrich (Yongin, South Korea).

### 4.2. Experimental Design

We used LDD175, NS1619, iberiotoxin, and paxilline as K^+^ channel modulators and employed computer-assisted sperm analysis to measure the motility and kinematic parameters of boar sperm. Sperm were treated with different concentrations of K^+^ channel modulators for 5 min. Relative membrane potential, intracellular calcium signals, and the pH of sperm were measured using time-lapse flowcytometry (TLFC), whilst acrosome-reacted sperm were estimated using classical flow cytometry. In TLFC, parameters were evaluated by accumulating a concentration of treatment/s in the setup. Fluorescence for each treatment/concentration was recorded for two minutes after the addition of the treatment/concentration. A detailed summary of concentrations and treatments is provided in the Appendix A. 

### 4.3. Sperm Medium

Sperm motility was measured in human tubal fluid (HTF) medium containing 101.6 mM of NaCl, 4.69 mM of KCl, 0.2 mM of Mg_2_SO_4_, 0.37 mM of KH_2_PO_4_, 2.04 mM of CaCl_2_, 25 mM of NaHCO_3_, 2.78 mM of glucose, 0.33 mM of Na-pyruvate, 21.4 mM of Na-lactate, and 0.4% (*w/v*) bovine serum albumin (BSA).

Non-capacitating medium (Non-CAP) included 135 mM of NaCl, 5 mM of KCl, 1 mM of MgSO_4_, 2 mM of CaCl_2_, 5 mM of glucose, 1 mM of sodium pyruvate, 10 mM of L-(+)-lactic acid, and 20 of mM 4-(2-hydroxyethyl) piperazine-1-ethane sulfonic acid (HEPES). Capacitating medium (CAP) consists of 15 mM of NaHCO_3_ and 4 g/L of BSA in the Non-CAP medium.

The calcium-free medium was made by replacing CaCl_2_ in the Non-CAP medium with 0.5 mM of ethylene glycol-bis(2-aminoethylether)-*N*,*N*,*N*′,*N*′-tetraacetic acid (EGTA). The pH values of all media were adjusted to 7.4 using 1M NaOH.

### 4.4. Sample Preparation

Commercially available boar seminal doses obtained from a local farm (Darby AI Center, South Korea) were used in this study, and all experiments strictly adhered to the guidelines of the Chungnam National University Animal Care Committee (201812A-CNU-01162). Seminal samples procured from diluted sperm-rich fractions were stored at 17 °C until the onset of experiments. All experiments were performed within 72 h of semen collection and screened for a minimum initial motility of 60%.

An approximately 10 mL semen sample was centrifuged at 600× *g* at 17 °C for 10 min, and the pellet containing sperm was washed out twice using the appropriate medium (Non-CAP or HTF). After washing, the sperm were resuspended in 5 mL of the medium and centrifuged at 600× *g* at 17 °C for 5 min followed by incubation at 37 °C and 5% CO_2_ for 1 h without disturbing the sperm pellet. After incubation, sperm were isolated from the liquid column (swim-up fraction), and were used for the experiments.

The motile-sperm isolated from the swim-up fraction was centrifuged at 600× *g* at 25 °C for 10 min, and the final concentration was adjusted to 1–2 × 10^7^ cells/mL in the appropriate medium. For capacitation, sperm resuspended in CAP medium were incubated at 37 °C and 5% CO_2_ for 4 h. For time-lapse flow cytometry data acquisition, the sperm concentration was adjusted to 1–3× 10^6^ cells/mL in the appropriate medium. Every experiment included a negative control counterpart without any treatment using the sperm of the same seminal dose as its treatment groups, processed under identical conditions. 

### 4.5. Computer-Assisted Sperm Analysis (CASA)

Sperm were concentrated to approximately 1 × 10^7^ cells/mL in HTF medium. Sperm motility and motion kinematics were analyzed using a computer-assisted sperm analysis (CASA) system (ISASv1^®^, Proiser S.L., Valencia, Spain), as described previously [62]. Briefly, 10 µL of the sperm sample was mounted on a pre-warmed Leja-counting slide (Cat# SC20-01-04-B; Nieuw Vennep, the Netherlands). Sperm were observed under 100× magnification in phase-contrast mode using an Olympus BX41 microscope (Olympus Europe GmbH, Hamburg, Germany). A minimum of 10 frames per treatment group with 100 sperm per field were acquired. After data acquisition, the digitized images were analyzed using ISAS. Sperm with a velocity of >5 µm/s were considered motile. The total motility was estimated using the following formula: total motile% = 100 − immotile sperms.

### 4.6. Membrane Potential Measurement

Sperm membrane potential changes were measured using the membrane-potential-sensitive fluorochrome: 3,3′-dipropyl thiadicarbo cyanine iodide (DISC_3_) (Cat#306, Invitrogen, Thermo Fisher Scientific, Waltham, MA 0251, USA). Temporal variation in membrane potential with different treatments was detected using a time-lapse flow cytometry setup, as explained in Matamoros-Volante et al. 2021 [63]. To validate this technique, we followed Matamoros-Volante’s protocol to generate standard curves using valinomycin and different potassium concentrations for theoretical membrane potential and fit the linear equation (Appendix A). One milliliter of sperm suspended in Non-CAP medium at a final concentration of 1–3 × 10^6^ cells/mL was loaded with 25 nM of DiSC_3_(5) and incubated at 37 °C and 5% CO_2_ for 15 min. Excess dye was washed via centrifugation at 600× *g* for 10 min, followed by resuspension in an equal volume of Non-CAP medium. A BD Accuri^TM^ C6 Plus flow cytometer (San Jose, CA, USA) with filter 3 (excitation: 488 nm, emission: 670LP) was employed for the time-lapse flow cytometry. Upon collecting the fluorescence data for the forward scatter (FSC) and side scatter (SSC), threshold limits were established to exclude cellular debris from sperm cells (gating #1), and only the cell population was gated. Of the gated cells, the data were spread against the forward scatter area (FSC-A) and the forward scatter height (FSC-H) to select singlets (gating #2). Data from the singlet population were recorded for time-lapse flow cytometry. Data acquisition was performed under 14 µL/min fluidics settings. Basal fluorescence measured during the first 2 min was followed by fluorescence recordings corresponding to different treatment concentrations added at 2 min intervals. The normalized median DiSC_3_ values were considered for comparisons between the different treatment groups. The equation (F_x_ − F_0_)/F_0_ was used for the data normalization of a sample, where F_x_ and F_0_ denote the fluorescence at a given point and fluorescence at a stable basal level, respectively [63]. 

### 4.7. Intracellular Calcium Measurement

An experimental setup for membrane potential measurements with a calcium-sensitive fluorochrome was used to detect intracellular calcium changes. Herein, 5 µM of 4-(6-Acetoxymethoxy-2,7-difluoro-3-oxo-9-xanthenyl)-4′-methyl-2,2′(ethylenedioxy)dianiline N,N,N′,N′-tetraacetic acid tetrakis (acetoxymethyl) ester: (Fluo4-AM) (Cat# F14201, Invitrogen, Thermo Fisher Scientific), together with 0.02% Pluronic^®^ F-127 (Cat# P3000MP, Invitrogen, Thermo Fisher Scientific), was added to 1 mL of sperm suspension in Non-CAP medium with the final concentration adjusted to 1–3 × 10^6^ cells/mL. Fluorochrome-loaded sperm were incubated for 30 min at 37 °C and 5% CO_2_ followed by the removal of excess dye via centrifugation at 600× *g* for 10 min and resuspension in an equal Non-CAP medium. To validate the calcium measurement system, 10 µM of ionomycin (calcium ionophore A23187, Cat# C7522, Sigma-Aldrich) and 5 mM of CaCl_2_ were added to the Fluo4-AM-loaded sperm sample after the first 2 min of basal fluorescence measurement (Appendix A). Sperm calcium signal changes were measured in both the Non-CAP (with external calcium) and calcium-free (without external calcium) media.

Samples were treated at 2 min intervals, and fluorescence changes were detected using filter 1 (excitation: 488 nm, emission: 533/30) of the BD Accuri^TM^ C6 Plus flow cytometer. Fluorescence changes were normalized against basal fluorescence values, and normalized values between the different treatment groups were compared.

### 4.8. Intracellular pH Measurement

In a time-lapse flow cytometry setup, intracellular pH changes were measured using pH-sensitive 2′, 7′,-bis-(2-carboxyethyl)-5-(and-6-) -carboxyfluorescein acetoxymethyl ester (BCECF-AM) dye. Briefly, motile sperm were mixed with 300 nM of BCECF-AM (Cat# B1170, Invitrogen, Thermo Fisher Scientific) and incubated at 37 °C and 5% CO_2_ for 10 min. After washing out the excess dye, fluorescence levels corresponding to the intracellular pH were measured using filter 1 (excitation: 488 nm, emission: 533/30). To identify the pH-responsive sperm subpopulation, we added 20 mM of NH_4_Cl, and the responding population was gated for the experiments (Appendix A). Different treatment concentrations were introduced at 2 min intervals.

### 4.9. Acrosome Reaction

The concentration of sperm isolated from the swim-up fraction was adjusted to 1–3 × 10^6^ cells/mL. Lectin of *Arachis hypogaea* (PNA) conjugated with the FITC method was employed to identify acrosome-reacted sperm in samples. Briefly, sperm were incubated at 37 °C and 5% CO_2_ for 4 h to induce capacitation in the CAP medium. Capacitated sperm were then treated with appropriate concentrations of LDD175 and NS1619 at 37 °C and 5% CO_2_ for 1 h. Treated samples were then washed out and loaded with 5 µg/mL of PNA-FITC (Cat#L7381, Sigma-Aldrich) and incubated for 15 min at 37 °C and 5% CO_2_. Excess dye was washed out via centrifugation, and sperm were then immobilized by adding 1 mL of 12.5% (*w/v*) paraformaldehyde in 0.5 M Tris (pH = 7.4), followed by incubation for 15 min at room temperature (21–25 °C). After immobilization, 3 µM of propidium iodide (Cat#P4170, Sigma-Aldrich) was added, immediately followed by FACS data acquisition. Under similar conditions, 10 µM of ionomycin was used to generate the positive control. Data for 10,000 cells/sample were recorded via flow cytometry. Dead sperm that were positive for PI staining (Filter 2: 585/40) were gated out using a PI-A vs. count histogram, and of the live cells that were negative for PI staining, FITC-positive cells (Filter 1: 533/30) that corresponded to acrosome-reacted sperm were counted. The acrosome-reacted sperm count was estimated as a percentage of the total number of counted cells.

### 4.10. Immunocytochemistry

Sperm immunocytochemistry was carried out as previously explained [64]. Briefly, sperm were fixed in 4% paraformaldehyde at room temperature (RT) for 20 min and then centrifuged at 1200× *g* for 6 min. The pellet was resuspended in 1x PBS (pH = 7.3). Sperm smears were made on poly-D-lysine-coated coverslips and allowed to dry at RT. Dry smears were washed out three times for 5 min each with PBS. Smears were then blocked with 5% BSA in PBS at 4 °C for 2 h. Smears were washed out again with PBS 3 times (5 min for each washout). Primary antibodies, KCNMA1 Rabbit pAb (Cat # A15283, ABclonal, Woburn, MA 01801, USA) and KCNU1 Rabbit pAb (Cat # A14967, ABclonal), were diluted to 1:100 in 5% BSA-PBS, and the washed smears were incubated with the diluted primary antibodies at 4 °C for overnight. After overnight incubation, smears were washed out with PBS 3 times and incubated with the secondary antibody, FITC Goat Anti-Rabbit IgG (H+L) (Cat# AS011, ABclonal), diluted to 1:200 in 5% BSA-PBS for 75 min at RT under dark conditions. Smears were thoroughly washed using PBS and then air-dried at RT before mounting with the permanent mounting reagent. Photographs of the immunolabeled sperm were obtained using confocal microscopy (Leica TCS SP8, Leica Biosystems Nussloch GmbH, Nußloch, Germany).

### 4.11. Apoptosis Assay

Sperm cell viability was measured using an ApoScreen^®^ Annexin V-FITC kit (Cat#10010-02, SouthernBiotech, Birmingham, AL, USA) with modifications. Sperm collected after swim-up was utilized for the assay. For the positive control, 3% formaldehyde in PBS was added to sperm and incubated on ice for 30 min. Sperm were then washed out with PBS, and fluorescence markers were added. To a volume of 100 µL of sperm suspension containing approx. 1 × 10^7^ cells/mL, 10 µL of Annexin V-FITC solution was added. The suspension was mixed well via gentle vortexing and incubated for 15 min on ice under light-protected conditions. After incubation, 350 µL of Annexin-binding buffer 1x diluted in distilled water was added, and 10 µL of 7-Aminoactinomycin D (7-AAD) (Cat# 10010-09, SouthernBiotech) was added immediately before flow cytometry data acquisition. Annexin V-FITC fluorescence data were acquired using filter 1 (488 nm/ 533/30 nm) of a BD Accuri C6 plus flow cytometer, while filter 3 (488 nm/ 670LP) was used to detect 7-AAD fluorescence. Unstained, single-stained with either Annexin V-FITC or 7-AAD, or double-stained (Annexin V-FITC and 7-AAD) samples of either the non-treated or positive control were used for color compensation. Sperm without the fluorescence (FITC-/7AAD-) were considered intact live cells, and their percentage values in different treatment groups were compared. The maximal effective concentrations (LDD175, 10µM; NS1619, 100 µM) and their negative control counterparts were assessed after 5, 30, and 60 min incubations. Ten thousand cells were analyzed for each sample; each treatment group consisted of five samples (n = 5).

### 4.12. Statistical Analysis

The results are presented as the mean ± S.D., and n refers to the number of trials performed. One-way ANOVA with Bonferroni correction (Origin Pro 8.1, Northampton, MA, USA) was used to calculate the statistical significance of data when comparing three or more datasets, while unpaired or paired Student’s *t*-tests were used to compare two groups. Differences were considered statistically significant at *p* < 0.05.

## 5. Conclusions

Because of the high tissue specificity and its participation in connecting sperm membrane potential and intracellular pH, which, in turn, initiates a complex cascade of signals, thereby opening a variety of other ion channels, a detailed understanding of the Slo3 channel in the context of different species is important for the study of fertilization. Although the role of K^+^ channels in a few species has been partially established, the principal K^+^ channel identity of many species remains largely speculative. Our study demonstrated that Slo3 is present in the boar sperm flagellar region. It is the principal channel participating in membrane potential modulation and is also involved in sperm motility, intracellular ion homeostasis, and acrosome reactions. Deciphering the identity of the ion channel contributing to the boar primary K^+^ conductance and its significance may help elucidate the boar sperm signaling cascade, which could aid the design of a novel generation of better sperm-handling procedures and improved fertility, and this would ultimately benefit animal breeding facilities and assisted reproductive technology.

## Figures and Tables

**Figure 1 ijms-24-07806-f001:**
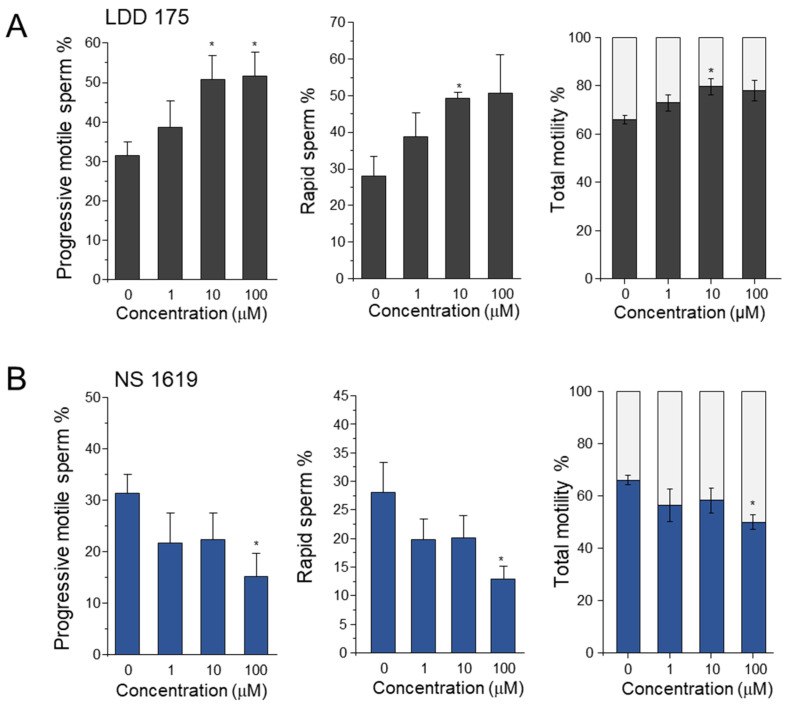
LDD175 treatment increased sperm motility parameters, while sperm treated with NS1619 exhibited a decreasing pattern. The left, middle, and right columns represent the progressive motility, rapid sperm%, and immotile sperm count, respectively. Row (**A**) depicts sperm subjected to the LDD175 treatment, while row (**B**) depicts sperm subjected to the NS1619 treatment. All samples were incubated for 5 min after treatment. Significant differences (*p* < 0.05) compared with the negative control with no treatment are marked with asterisks (n = 10, average ± S.D.).

**Figure 2 ijms-24-07806-f002:**
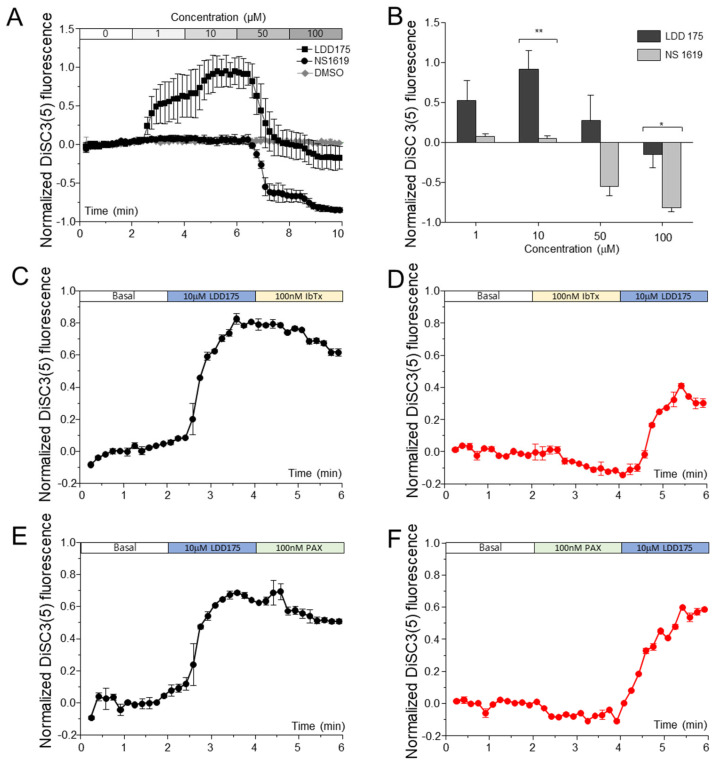
The membrane potential of sperm is differently modulated by LDD175 and NS1619. Panel A depicts the relative changes in membrane potential signals upon treatment with LDD175 (■) and NS1619 (●) compared with the DMSO vehicle control (◆) (**A**). The bar graph (**B**) compares the relative signals at each concentration after the fluorescence reaches a stable point. (**C**–**F**) represent changes in membrane potential corresponding to DiSC3(5) fluorescence upon the sequential treatment of LDD175 with IbTx (**C**,**D**) and PAX (**E**,**F**). Data for each concentration contain recordings for 2 min and are presented as the average ± S.D. (n = 3, * *p* < 0.05, ** *p* < 0.01). Changes in membrane potential with Slo1-specific blockers (PAX and IbTx) are shown in the Appendix A.

**Figure 3 ijms-24-07806-f003:**
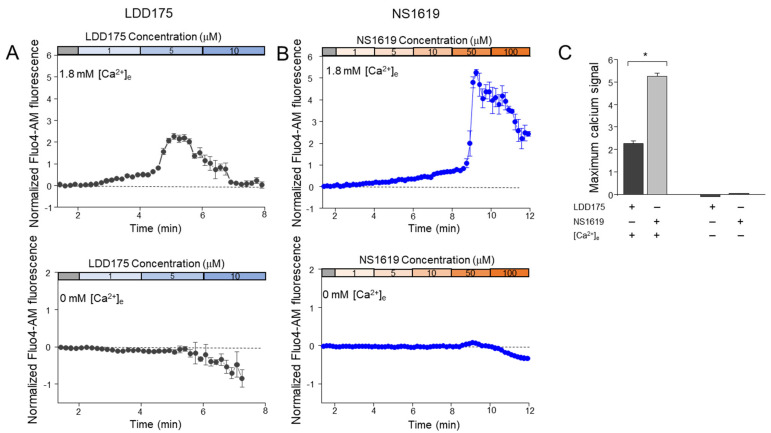
Both the LDD175 and NS1619 treatments increase the cytosolic calcium of boar sperm in an extracellular−calcium−dependent way. The upper row shows calcium signal changes in 1.8 mM of calcium medium, whereas the bottom row shows the corresponding signals in the Ca^2+^−free medium. Signals in LDD175− (**A**) and NS1619− (**B**) treated sperm are dark gray and blue, respectively. Data are presented as the average ± S.D. The bar graph (**C**) compares the maximum signals in 1.8 mM of Ca^2+^−medium (5 μM of LDD175, 50 μM of NS1619) and its counterparts corresponding to the same concentrations in Ca^2+^-free medium (n = 3, * *p* < 0.05).

**Figure 4 ijms-24-07806-f004:**
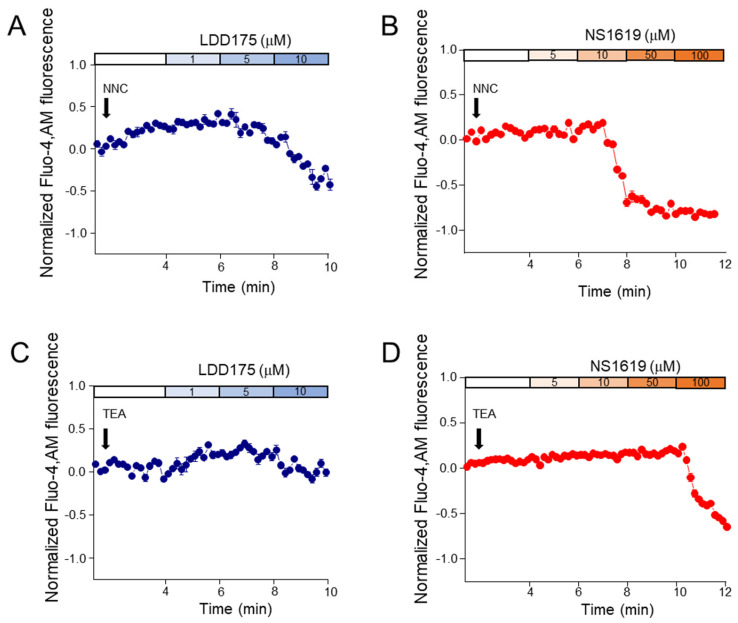
CatSper and K^+^ channel blockers reduce intracellular calcium signals. The top row shows (**A**,**B**) changes in Fluo4−AM signals after treatment with 1 μM of NNC55−0396 preceded by K^+^ channel modulators. Signal changes after treatment with 60 mM of TEA followed by the respective K^+^ channel modulators are illustrated in the bottom row (**C**,**D**). Blue traces represent [Ca^2+^]_i_ changes after LDD175 treatment, while red traces represent changes after NS1619 treatment. The data for each LDD175 and NS1619 treatment had an acquisition duration of 2 min (n = 3).

**Figure 5 ijms-24-07806-f005:**
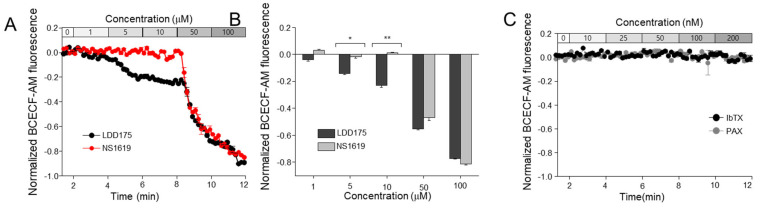
LDD175 and NS1619 acidify boar spermatozoa. This figure shows the changes in intracellular pH after treatment with LDD175 and NS1619. (**A**) shows the traces of normalized fluorescence during the data acquisition, with light red indicating the signals of the NS1619-treated group and black indicating the signals of the LDD175−treated group. The fluorescence of each concentration of LDD175 and NS1619 is compared in the bar graph (**B**). The effects of IbTx and PAX on sperm pH are shown in Figure (**C**). Data are presented as the average ± S.D. (n = 3, * *p* < 0.05, ** *p* < 0.01).

**Figure 6 ijms-24-07806-f006:**
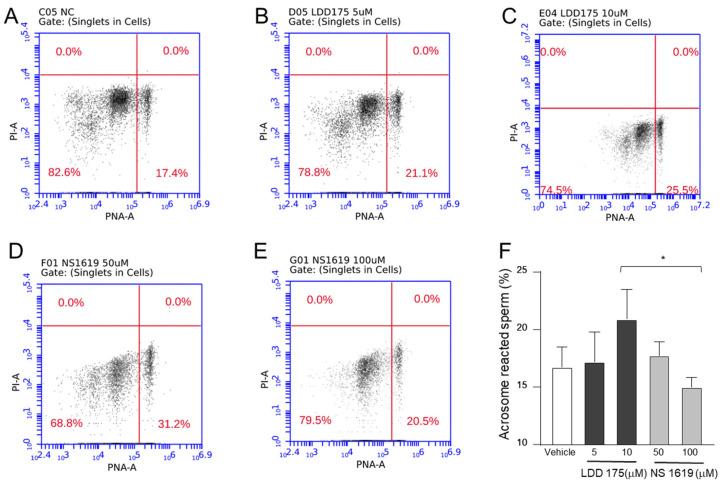
Activation of Slo3 increases the number of acrosome-reacted boar sperm. (**A**–**E**) show representative quadrant plots of the negative control (**A**), LDD175 ((**B**): 5 µM; (**C**): 10 µM), and NS1619 ((**D**): 50 µM; (**E**): 100 µM). The bar graph shows (**F**) the percentage of acrosome-reacted sperm cells in each treatment condition, where white, dark gray, and light gray boxes represent the negative control, the LDD175 treatment, and the NS1619 treatment, respectively. Data are presented as the average ± S.D. (n = 5, * *p* < 0.05). Single staining conditions containing only PNA (Appendix A) and PI (Appendix A) that were used to obtain the necessary gating and the positive control containing 10 µM ionomycin (Appendix A) are provided in the Appendix A.

**Figure 7 ijms-24-07806-f007:**
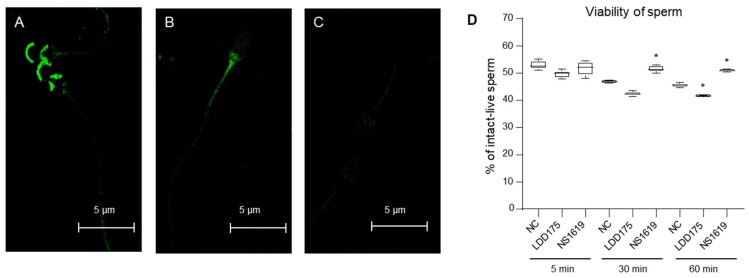
Localization of Slo channels in immunocytochemistry and boar sperm viability. (**A**) Anti-Slo1 antibody; (**B**) anti-Slo3 antibody; (**C**) control experiment using only the secondary antibody. White scale bar (**A**–**C**) represents 5 µm. (**D**) Percentage of intact live cells obtained from an apoptosis assay (n = 5, average ± S.D; * *p* <0.05).

## Data Availability

Data are contained within the article or Appendix A.

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
