# Peer review of "Pharmacological Evidence Suggests That Slo3 Channel Is the Principal K+ Channel in Boar Spermatozoa"

_ijms, 2023, doi:10.3390/ijms24097806_

Round 1
Reviewer 1 Report (Previous Reviewer 2)
The reviewers' comments have been addressed. The manuscript is ready for publication in IJSM.
Author Response
Thank you for taking your time to review our manuscript.
Reviewer 2 Report (New Reviewer)
I have several minor suggestions. I listed them with line numbers.
L210: It is not clear whether it is also part of the figure legend.
L229: It is not clear whether it is also part of the figure legend.
L450: time-lapse flow cytometry _ Abbreviation is needed.
L736: Format of the citations is not consistent, including the number of the authors, use of the capital letters for title of the manuscript, journal name abbreviation and so on.
L766: It appears that there are errors in this citation.
Author Response
We appreciate your time and effort put into reviewing and improving the quality of our manuscript. We revised our manuscript incorporating your comments and suggestions. The following are our point-by-point responses to each of your comments.
- L210 and L229: It is not clear whether it is also part of the figure legend.
Thank you for your valuable comment to improve the quality of the manuscript. We have adjusted the formatting settings to avoid confusion.
- L450: time-lapse flow cytometry _ Abbreviation is needed.
Thank you for your comment on improving the readability of the manuscript. We have defined the abbreviation and also included it under the section “Abbreviation”.
- L736: Format of the citations is not consistent, including the number of the authors, use of the capital letters for title of the manuscript, journal name abbreviation and so on.
L766: It appears that there are errors in this citation.
Thank you very much for your comment. We have carefully reviewed the bibliography and edited it to provide a clear citation list with a consistent format.
Reviewer 3 Report (New Reviewer)
In this Manuscript the authors use pharmacological methodology to determine that Slo3 channel is the principal K+ channel in boar spermatozoa, with relevance for porcine sperm function. This is potentially interesting, although purely pharmacological work has been known to provide unspecific information and there are a few issues the authors really need to address.
- The authors need to provide evidence (WB and ICC) of the presence of Slo channels on porcine sperm, notably Slo3. I have not found such information in the literature, and without it this data loses potential impact.
- The concentrations of inhibitors used are very high (especially those that cause major effects). Although there are mentions of appropriate vehicle (DMSO) controls, the authors need to provide parallel evidence that boar sperm viability is not compromised under their experimental conditions. The easiest way for a sperm to lose motility, for example, is that it is compromised and no longer viable, and this has been the case in many similar studies.
I emphasize that the controls mentioned above for sperm viability must be done for all relevant experimental conditions, not just some of them.
-The relevant inhibitors used are in the micromolar range, while others, that show no effect and are discussed as controls (PAX, ibTX), are in the nanomolar range. This is a huge difference and it needs to be discussed more, and higher doses of these compounds used to provide appropriate comparisons.
-Besides acrosome reaction the authors need to monitor capacitation as well in the presence/absence of Slo3 inhibition.
Author Response
Thank you for taking time to review our manuscript. We are truly grateful for your feedback. We have considered your thoughtful comments and made the corrections and clarifications accordingly. Our point-by-point responses to each of your comments are as follows.
- The authors need to provide evidence (WB and ICC) of the presence of Slo channels on porcine sperm, notably Slo3. I have not found such information in the literature, and without it this data loses potential impact.
Answer: Thank you for your comment on making the manuscript scientifically sounder. We carried out sperm immunocytochemistry to address this and updated the following. Additionally, data obtained from the new experiment are presented in Figure 7 (A-C).
Results: L238- 254:
2.6. Slo1 Slo3 channels showed different localizations on boar spermatozoa.
Immunofluorescence studies show that most Slo1 channels were present in the head area with two concentrated regions, the anterior head region, and the post-acrosomal region (Figure 7 A). Slo3 channel fluorescence signals were less strong than Slo1 under the same confocal microscopic settings. However, a distinct area with apparent green fluorescence depicting Slo3 channel was seen in the proximal tail area, and such fluorescence was absent in its head area (Figure 7 B). Non-specific binding of the secondary antibody was not detected (Figure 7 C).
Discussion: L 451: 462
The present study through immunolocalization shows that Slo3 channel of boar sperm is mainly located in the flagellum. Sperm tail is significant in its motility control and majority of the motility-related ion channels are generally found in the principal piece of sperm flagellum (Sun et al., 2017; Vicente-Carrillo et al., 2017; Wang et al., 2021). Additionally, most of these motility-associated channels are directly controlled by changes of K+ conductance through sperm transmembrane. The physical proximity of Slo3 channel to sperm principal piece corroborates its functional relevance to sperm motility and associated signaling pathways. Marc et. al. showed the presence of Slo1 channel throughout the flagellum and anterior area of the post-acrosomal region of boar sperm (Yeste et al., 2019). However, we found Slo1 to be only localized in the head area. Furthermore, we also found Slo1 in sperm apex area closer to acrosome reaction. This could be explained by the significance of Slo1 in acrosome reaction as evident from literature (Yeste et al., 2019).
Materials and Methods: L631-647
- Immunocytochemistry
Sperm immunocytochemistry was carried out as previously explained (Álvarez-rodriguez et al., 2018). Briefly, sperm were fixed in 4% paraformaldehyde at room temperature (RT) for 20 mins and then centrifuged at 1,200 × g for 6 mins. The pellet was resuspended in 1x PBS (pH = 7.3). Sperm smears were made on poly-D-lysine coated coverslips and allowed to dry at RT. Dry smears were washed out three times for 5mins each with PBS. Smears were then blocked with 5% BSA in PBS at 4°C for 2 hours. Smears were washed out again with PBS for 3times (5min for each washout). Primary antibodies: KCNMA1 Rabbit pAb (Cat # A15283, ABclonal, MA 01801, USA) and KCNU1 Rabbit pAb (Cat # A14967, ABclonal) were diluted to 1:100 in 5% BSA-PBS, and the washed smears were incubated with the diluted primary antibodies at 4°C for overnight. After overnight incubation, smears were washed out with PBS 3 times and incubated with the secondary antibody: FITC Goat Anti-Rabbit IgG (H+L) (Cat# AS011, ABclonal), diluted to 1:200 in 5% BSA-PBS for 75 minutes at RT under dark conditions. Smears were thoroughly washed using PBS and then air dried at RT before mounting with the permanent mounting reagent. Photographs of the immunolabeled sperm were obtained using confocal microscopy (Leica TCS SP8, Leica Biosystems Nussloch GmbH, Germany).
- The concentrations of inhibitors used are very high (especially those that cause major effects). Although there are mentions of appropriate vehicle (DMSO) controls, the authors need to provide parallel evidence that boar sperm viability is not compromised under their experimental conditions. The easiest way for a sperm to lose motility, for example, is that it is compromised and no longer viable, and this has been the case in many similar studies. I emphasize that the controls mentioned above for sperm viability must be done for all relevant experimental conditions, not just some of them.
Answer: We highly appreciate your comment. We agree with the reviewer, and we tested the highest effective concentrations of the treatments used in an apoptosis assay to delineate possible effects on sperm viability. Findings of the experiment are summarized in Figure 7D and we have included the following in the manuscript.
Results: L247-256
2.7 Slo3 channel inhibition improves boar cell viability.
After 5min of incubation with LDD175 (10 µM) and NS1619 (100 µM), the intact live sperm population of both treatment groups was not significantly different from that of the negative control, and the live cell population of NC, LDD175, and NS1619 were 52.9%, 49.5%, and 51.7%, respectively (Figure 7 D). After 30min incubation, the values of NC and LDD175 slightly decreased while NS1619 remained unchanged compared to their 5 min-incubated counterparts. Of 30min-incubated samples, the intact-live cell population of 100µM NS1619 was significantly higher than its negative control. After one hour of incubation with the treatments, LDD175 resulted in the lowest intact-live cells, while the highest was with NS1619, significantly different from the negative control (n=5, p<0.05).
Discussion: L463-472
To confirm if the decreased motility coincided with decreased sperm viability, we assessed sperm viability using an apoptosis assay. We used the maximal effective concentration of LDD175 and NS1619, and treated sperm were incubated for 5, 30, and 60mins. For 5min-incubation, the intact-live cell population was not significantly different from that of the negative control. However, as the incubation period increased, we observed that inhibition of Slo3 can improve sperm membrane intactness, while Slo3 activated-sperm showed the least population of intact-live cells. This observation is consistent with the findings of Jean et al., who reported that inhibition of Slo3 channel in mouse and bull sperm can extend their viability (Jean et al., 2018). Thus, it is safe to hypothesize that the motility reduction of sperm is not due to compromising its viability.
Materials and Methods: L649-669
- Apoptosis assay
Sperm cell viability was measured using ApoScreen® Annexin V-FITC kit (Cat#10010-02, SouthernBiotech, Birmingham, AL 35209, USA) with modifications. Sperm collected after swim-up was utilized for the assay. For the positive control, 3% formaldehyde in PBS was added to sperm and incubated on ice for 30mins. Sperm were then washed out with PBS, and fluorescence markers were added. To a volume of 100 µL of sperm suspension containing approx. 1 × 107 cells/mL, 10 µL of Annexin V-FITC solution was added. The suspension was mixed well by gentle vortexing and incubated for 15mins on ice under light-protected conditions. After incubation, 350 µL of Annexin binding buffer 1x diluted in distilled water was added, and 10 µL of 7-Aminoactinomycin D (7-AAD) (Cat# 10010-09, SouthernBiotech) was added immediately before flow cytometry data acquisition. Annexin V-FITC fluorescence data was acquired using filter 1 (488nm/ 533/30 nm) of BD Accuri C6 plus flow cytometer, while filter 3 (488 nm/ 670LP) was used to detect 7-AAD fluorescence. Unstained, single stained with either Annexin V-FITC or 7-AAD, double stained (Annexin V-FITC and 7-AAD) samples of either non-treated or positive control were used for color compensation. Sperm without both the fluorescence (FITC-/7AAD-) were considered intact-live cells, and its percentage values among different treatment groups were compared. The maximal effective concentrations (LDD175 10µM, NS1619 100 µM) and their negative control counterparts were assessed after 5-, 30-, and 60-min incubations. Ten thousand cells were analyzed for each sample; each treatment group consisted of five samples (n=5).
- The relevant inhibitors used are in the micromolar range, while others, that show no effect and are discussed as controls (PAX, ibTX), are in the nanomolar range. This is a huge difference and it needs to be discussed more, and higher doses of these compounds used to provide appropriate comparisons.
Answer: Thank you for your comment. The concentrations used are based on previous literature. Iberiotoxin and paxilline are well-established compounds that inhibit Slo1 current at 100nM concentration (Mannowetz et al., 2013; Noto et al., 2021; Sánchez-Carranza et al., 2015; Yeste et al., 2019).Additionally, according to Tharaka et al. the effective concentration of LDD175 and NS1619 with electrophysiological recording are 10 µM and 100 µM (Wijerathne et al., 2017). Based on this background, we tested the concentration range of each treatment including its effective concentration (Max. ef). As there were no obvious effects with IbTx and PAX at 100nM, maximal effective concentration, we even increased the concentration up to twice its Max. ef. (200 nM). We believe that these different concentrations of the treatment are due to different action mechanisms which are not fully elucidated yet.
We included the following under the “Discussion” to address this.
L315-318: These data coincide with the concentrations corresponding to maximum Slo3 activation upon LDD175 treatment and inhibition upon NS1619 treatment in the patch clamping recording, which were 10 µM and 100 µM, respectively (Wijerathne et al., 2017).
L320-324: To further confirm the role of Slo3 in maintaining boar sperm membrane potential, we also assessed membrane potential changes with Slo1-specific blockers such as IbTx and PAX. According to the literature, Slo1 inhibiting concentrations of both chemicals are approximately 100 nM (Mannowetz et al., 2013; Tang et al., 2010). The membrane potential in groups treated with up to 200 nM of both chemicals remained relatively unchanged.
- Besides acrosome reaction the authors need to monitor capacitation as well in the presence/absence of Slo3 inhibition
Answer: We are grateful for your comment. As the study focuses on acute treatment and transient changes of ion concentrations across the sperm membrane, we omit to monitor capacitation in this study. Furthermore, the literature provides evidence that membrane hyperpolarization itself is enough to complete the acrosome reaction in a capacitation-independent manner (De La Vega-Beltran et al., 2012). Therefore, in our study, in addition to intracellular ionic measurement, we also measured spontaneous acrosome reaction.
Round 2
Reviewer 3 Report (New Reviewer)
The authors have adequately addressed my concerns. I have no further comments
This manuscript is a resubmission of an earlier submission. The following is a list of the peer review reports and author responses from that submission.
Round 1
Reviewer 1 Report
The manuscript: Pharmacological evidence suggests Slo3 channel to be the principal KSper in boar spermatozoa by Akila Cooray and colleagues uses a pharmacological approach to identify the molecular identity responsible of potassium currents in boar sperm. This is a very interesting and important subject due to role of membrane potential in sperm physiology and the differences and controversy about the presence of Slo3 vs Slo1 in human sperm, as opposed to apparently the sole presence of Slo3 in mouse sperm.
However, I have important concerns about this manuscript:
1. Throughout the paper the term Ksper is used as if it was a channel. Ksper refers to potassium currents in sperm that may be produced by one or more molecular entities. Therefore it can be inhibited (pharmacologically) but not deleted (genetically) as stated in the manuscript. The molecular entity(ies) responsible for KSper current is not fully elucidated.
2. The authors talk about K+ extrusion, ion channels can work in both directions (except for Hv1), therefore that term should be avoided. The direction of ion movement depends on the electrochemical gradient.
3. Slo3 is believed to be expressed only in sperm. However there are some reports indicating it may be found elsewhere (Chavez et al., 2019).
4. Human Slo3 regulation is open to debate, some argue is calcium and others pH or maybe both, subtle affirmations regarding this notion are recommended.
5. It is unclear what is meant by: Slo3 of human spermatozoa also play a non-redundant role.
6. The main deficiency of this paper is that all the conclusions are based on pharmacological data, and it is very difficult to claim that inhibitors are 100% specific for a particular ion channel. There is always the possibility of cross effect, and this is not mentioned at all.
7. The use of fluorescent dyes to measure calcium, pH and membrane potential (Em) in sperm cells is very well established and accepted. However, in this paper necessary controls and calibrations were not performed. The Em determinations require an internal calibration for each trace using valinomycin and potassium additions. That is the beauty of the method, quantitative values of Em can be determined making the results comparable. This dye partitions in the membrane and this can vary from experiment to experiment. Therefore, this dye should not be washed as mentioned in Methods. As opposed to calcium and pH dyes that need to be loaded into the cell and excess dye must be washed. For calcium measurements it is mentioned that ionomycin and Mn are used, but this is not shown. The same for pH experiments, the controls are not shown.
8. It is not clear how the data is normalized.
9. It is not clear how the populations are gated, especially is difficult to believe that they have 0% of dead cells. How many events are analyzed per condition?
10. It is not well explained the rationale for the use of some compounds. NNC for example.
11. What is the positive control of AR to compare with?
12. What is the rationale of membrane hyperpolarization to control sperm motility? The references given do not correspond.
13. The title of the figures should be the conclusion of the experiment, as opposed to: Effect of......
14. English must be revised
Author Response
Reviewer #1
Thank you for taking time to review our manuscript. We are truly grateful for your feedback. We have considered your thoughtful comments and made the corrections and clarifications accordingly. Our point-by-point responses to each of your comments as follows.
- Throughout the paper the term Ksper is used as if it was a channel. Ksper refers to potassium currents in sperm that may be produced by one or more molecular entities. Therefore, it can be inhibited (pharmacologically) but not deleted (genetically) as stated in the manuscript. The molecular entity(ies) responsible for KSper current is not fully elucidated.
Response:Thank you for your thoughtful comment in helping us use accurate terminology. We have revised the manuscript accordingly. The corrections made are as follows.
Line 13-15: “We aimed to determine the channel identity of boar sperm contributing to the KSper using pharmacological dissection.”
Line 46-48: “Deletion or inhibition of the ion channels contributing to KSper in spermatozoa negatively affects their ability to complete fertilization [7]–[9].”
- The authors talk about K+ extrusion, ion channels can work in both directions (except for Hv1), therefore that term should be avoided. The direction of ion movement depends on the electrochemical gradient.
Response:Thank you for your valuable comment. We have corrected the terminology throughout the manuscript to avoid confusion and increase its readability.
Previously written “K+ extrusion” in line 11, 50 was re-worded to “K+ flux”
- Slo3 is believed to be expressed only in sperm. However, there are some reports indicating it may be found elsewhere (Chavez et al., 2019).
Response:Thank you for your comment on making our manuscript scientifically sounder and more updated.We have updated lines 57- 59 in the manuscript to incorporate the changes and the additional reference corresponding to the new information was included.
Line 53-55: “Slo3 is mainly expressed in mammalian male reproductive tissues[10]; however, a recent study by Chávez et al. found cytoplasmic Slo3 expression in somatic tissues [11].”
- Human Slo3 regulation is open to debate, some argue is calcium and others pH or maybe both, subtle affirmations regarding this notion are recommended.
Response:We highly appreciate your valuable recommendation. We agree with the reviewer and have revised our manuscript to include the suggested notion. We have included the following clause (Line 58-60) and sited the appropriate references (Brenker et al., 2014; Leonetti et al., 2012; Wijerathne et al., 2019; Zhang et al., 2006).
Line 58-60:“However, the precise mechanism underlying the regulation of human Slo3 remains un-clear. Literature supports the fact that Slo3 is gated by intracellular pH, intracellular cal-cium, or both [15], [17]–[20].”
- It is unclear what is meant by: Slo3 of human spermatozoa also play a non-redundant role.
Response:Thank you very much for your comment. We have edited our manuscript elaborating the mentioned sentence-fragment to provide a more complete overview and avoid confusion. We have also included a new reference to support this claim.
Line 63-65: “Slo3 in human spermatozoa also plays a vital role in sperm physiology, as homozygous mutations in human Slo3 result in severe asthenozoospermia and selective inhibition impairs sperm function [18], [21], [22].”
- The main deficiency of this paper is that all the conclusions are based on pharmacological data, and it is very difficult to claim that inhibitors are 100% specific for a particular ion channel. There is always the possibility of cross effect, and this is not mentioned at all.
Response:We thank you for your thoughtful comment to make our manuscript complete. We agree with the reviewer, and we have included the following clause into the discussion of the manuscript to address the mentioned short-coming.
Line 411-414: “Furthermore, the current study utilizes pharmacological data, which may have been con-voluted by the cross-reactivity of chemicals with different ion channels. Future patch-clamping experiments using chemicals with ion channels of interest will increase the reliability of this data.”
- The use of fluorescent dyes to measure calcium, pH, and membrane potential (Em) in sperm cells is very well established and accepted. However, in this paper necessary controls and calibrations were not performed. The Em determinations require an internal calibration for each trace using valinomycin and potassium additions. That is the beauty of the method, quantitative values of Em can be determined making the results comparable. This dye partitions in the membrane and this can vary from experiment to experiment. Therefore, this dye should not be washed as mentioned in Methods. As opposed to calcium and pH dyes that need to be loaded into the cell and excess dye must be washed. For calcium measurements it is mentioned that ionomycin and Mn are used, but this is not shown. The same for pH experiments, the controls are not shown.
Response: We thank you for your insightful comment regarding technical details. Our experiment, modified Matamoros-Volante's protocol to measure membrane potential and calcium levels separately. To achieve this, we measured standard curves using valinomycin and different potassium concentrations (as shown in the supplementary figure 3.). Although the dye was washed out for membrane potential measurements, we validated our results by comparing them with test results that did not wash out the dye (below figures A and B). The effects of LDD175 and NS1619 were consistent with our condition. Due to limited sample size and variations between trials, we expressed the change in membrane potential as a relative value to the measured fluorescence intensity. Fluo-4 and BCECF fluorescence measurements were also tested with 10mM ionomycin and 20mM NH4Cl as control experiments (Supplementary figure 4.). Each control experiment is provided in the supplementary figures.
we have included the following clause into the materials and methods of the manuscript..
Line 533-536: To validate this technique, we followed Matamoros-Volante's protocol to generate stand-ard curves using valinomycin and different potassium concentrations for theoretical membrane potential and fit the linear equation to validate (Supplementary figure 3).
- It is not clear how the data is normalized.
Response:We are thankful for your comment in improving the clarity of our manuscript. We have included the following including the equation we used to show how the data normalization was done.
Line 551-554: “The equation (Fx-F0)/F0 was used for the data normalization of a sample where Fx and F0 denote the fluorescence at a given point and fluorescence at a stable basal level, respec-tively [55].”
- It is not clear how the populations are gated, especially is difficult to believe that they have 0% of dead cells. How many events are analyzed per condition?
Response:Thank you for your comment. For TLFC, time period was used as the limit and initial sperm concentration in the suspension and the fluidics kept unchanged throughout the experiment. For acrosome reaction, 10000 of singlets were collected. To further elaborate and clarify the methods we have included/amended the following into the revised manuscript.
Line 489-495: “Upon collecting the fluorescence data for forward scatter (FSC) and side scatter (SSC), threshold limits were established to exclude cellular debris from sperm cells (gating #1) and only the cell population was gated. Of the gated cells, the data was spread against forward scatter area(FSC-A) and forward scatter height (FSC-H) to select the singlets (gat-ing #2). Data from the singlet population was recorded for time-lapse flow cytometry. Data acquisition was performed under 14 µL/min fluidics settings.”
Line 545-549: “Dead sperm that were positive for PI staining (Filter 2: 585/40) were gated out using a PI-A vs. count histogram, and of the live cells that were negative for PI staining, FITC-positive cells (Filter 1: 533/30) that corresponded to acrosome-reacted sperm were counted. The acrosome-reacted sperm count was estimated as a percentage of the total number of counted cells.”
- It is not well explained the rationale for the use of some compounds. NNC for example.
Response:We thank you very much for your valuable comment. We have included the following to elaborate on the rationale for using NNC 55-0396.
Line 335-339: “. We postulated that if the membrane potential of the sperm affect calcium permeability through CatSper channel, calcium elevation of CatSper blocker pre-treated sperm upon LDD175 or NS1619 treatment should be less than that without the CatSper blocker, there-fore, CatSper inhibitor-treated sperm were treated with LDD175 and NS1619 separately.”
- What is the positive control of AR to compare with?
Response:Thank you for your comment. 10 µM ionomycin treated under similar conditions as its treatment counterparts was used as a positive control for the experiment (Supplementary figure 2). We also added the following under the results section to avoid further confusions.
Line 224-225: “The a crosome-reacted sperm counts of all treatment groups were lower than that of the positive control (supplementary Figure 2).”
- What is the rationale of membrane hyperpolarization to control sperm motility? The references given do not correspond.
Response:Thank you for your valuable comment. We apologize for the error occurred in referencing. We have carefully reviewed the manuscript again and amended the corresponding references.
- The title of the figures should be the conclusion of the experiment, as opposed to: Effect of......
Response:Thank you very much for your comment to bring our manuscript to the standards. We have renamed all the figure titles (figure 1-6) highlighting the conclusion of the individual experiment or the data set. The renamed figure titles are as follows.
Figure 1. LDD175 treatment increases sperm motility parameters while sperm treated with NS1619 exhibited a decreasing pattern.
Figure 2. The membrane potential of sperm is differently modulated by LDD175 and NS1619
Figure 3. Both LDD175 and NS1619 treatments increase the cytosolic calcium of boar sperm in an extracellular calcium-dependent way.
Figure 4. CatSper and K+ channel blockers reduce intracellular calcium signals.
Figure 5. LDD175 and NS1619 acidify boar spermatozoa.
Figure 6. Activation of Slo3 increases the number of acrosome-reacted boar sperm.
- English must be revised
Response:Thank you for your valuable input to improve the quality of our manuscript. We have thoroughly reviewed our manuscript and corrected English where necessary. We also received assistance from a professional English editing service from Editage.com to review the language of the manuscript.

Reviewer 2 Report
Comments to the manuscript “Pharmacological evidence suggests Slo3 channel to be the principal KSper in boar spermatozoa”
The manuscript is original and well-written. It was demonstrated that Slo3 channel is the principal KSper in boar spermatozoa and the relevance of Slo3 in sperm membrane potential, motility, [Ca2+]i, pH, and acrosome reaction.
MAJOR COMMENTS
-Line 91, “Total motility of sperm increases with LDD175 up to nearly 80% when incubated for 5mins. The motility increase with LDD175 is concentration dependent with the maximum motility observed with 10 μM LDD175 and thereafter the motility decreases to basal level with higher concentrations”. Please note that these results of total motility are not shown in Fig. 1. If the total motility was calculated by the sum of progressive sperm and rapid sperm, the maximum motility in LDD175 is higher than 90% at 10 μM, and no motility decrease is observed at the higher concentration used (100 μM). Finally, the sum of progressive, rapid and immotile sperm is higher than 100% in LDD175 10 and 100 μM groups. Please double check and correct if necessary.
-Line 343: It is intriguing that Slo3 activation by LDD175 promoted several sperm capacitation hallmarks, such as increased membrane hyperpolarization, motility, and intracellular levels of calcium, but did not caused the intracellular alkalinization characteristic of sperm capacitation. On the contrary, the pH of sperm became acidic with the increasing LDD175 concentrations. Please discuss in this regard.
-The role of Slo3 in boar sperm spontaneous acrosome reaction was analyzed in this study. Please discuss if similar results would be obtained if the induced acrosome reaction was analyzed.
-The discussion could be enriched with results of the following articles:
Lyon M et al. A selective inhibitor of the sperm-specific potassium channel SLO3 impairs human sperm function. Proc Natl Acad Sci U S A. 2023 Jan 24;120(4):e2212338120. doi: 10.1073/pnas.2212338120
Escoffier J et al. Flow cytometry analysis reveals that only a subpopulation of mouse sperm undergoes hyperpolarization during capacitation. Biol Reprod. 2015 May;92(5):121. doi: 10.1095/biolreprod.114.127266
Abi Nahed R et al. Slo3 K+ channel blocker clofilium extends bull and mouse sperm-fertilizing competence. Reproduction. 2018 Dec 1;156(6):463-476. doi: 10.1530/REP-18-0075 (This article could be of use to enhance one of the possible applications of the study)
MINOR COMMENTS
Line 46 Please replace “Capacitated sperm” with “In capacitated sperm”
Line 55, please consider replacing “homologous” with homology
Author Response
Reviewer #2
We appreciate your time and effort put into reviewing and improving the quality of our manuscript. We revised our manuscript incorporating your thoughtful comments and suggestions. Followings are our point-by-point responses to each of your comments.
MAJOR COMMENTS
- Line 91, “Total motility of sperm increases with LDD175 up to nearly 80% when incubated for 5mins. The motility increase with LDD175 is concentration dependent with the maximum motility observed with 10 μM LDD175 and thereafter the motility decreases to basal level with higher concentrations”. Please note that these results of total motility are not shown in Fig. 1. If the total motility was calculated by the sum of progressive sperm and rapid sperm, the maximum motility in LDD175 is higher than 90% at 10 μM, and no motility decrease is observed at the higher concentration used (100 μM). Finally, the sum of progressive, rapid, and immotile sperm is higher than 100% in LDD175 10 and 100 μM groups. Please double check and correct if necessary.
Response:We greatly appreciate your valuable comment in delivering clearer results. We estimated the total motility of the treatment groups using [Total motility% = 100- Immotile %]. Furthermore, this value was equal to the sum of progressive motile % and non-progressive motile % (not shown in the manuscript). As suggested, to avoid readers’ confusions and increase readability, we have included the equation in line 104 and 472.
- Line 343: It is intriguing that Slo3 activation by LDD175 promoted several sperm capacitation hallmarks, such as increased membrane hyperpolarization, motility, and intracellular levels of calcium, but did not cause the intracellular alkalinization characteristic of sperm capacitation. On the contrary, the pH of sperm became acidic with the increasing LDD175 concentrations. Please discuss in this regard.
Response:We thank you for your thoughtful comment to help us provide a more complete overview for the manuscript. We have added the following to incorporate your suggestion.
Line: 369-374 “Acidification upon LDD175 treatment could be due to the inhibition of Hv1 by membrane hyperpolarization, as boar sperm is found to have functional Hv1 channels that aid in proton (H+) efflux from sperm, sperm membrane depolarization is needed for active func-tioning of the Hv1 channel; furthermore, in our initial experiments, we found that LDD175 can hyperpolarize the sperm plasma membrane.”
- The role of Slo3 in boar sperm spontaneous acrosome reaction was analyzed in this study. Please discuss if similar results would be obtained if the induced acrosome reaction was analyzed.
Response:We highly appreciate your comment. We have included the following under the discussion to further elaborate our postulation regarding the situation.
Line 396-397: “In our study, we estimated the spontaneous acrosome reaction in capacitating medium.”
Line 401-407: “However, it is noteworthy that the difference in the studies is that previous experiments measured progesterone-induced acrosome exocytosis (10µg/ml) whereas such an acro-some reaction induction was not employed in our study. Given that membrane hyperpo-larization is itself sufficient [36], [51] and a key regulatory factor for the completion of acrosome reaction, we deduce the trends observed regarding the acrosome reaction in the study would be the same but at a higher magnitude under acrosome-exocytosis inducing conditions.”
- The discussion could be enriched with results of the following articles:
Lyon M et al. A selective inhibitor of the sperm-specific potassium channel SLO3 impairs human sperm function. Proc Natl Acad Sci U S A. 2023 Jan 24;120(4):e2212338120. doi: 10.1073/pnas.2212338120
Escoffier J et al. Flow cytometry analysis reveals that only a subpopulation of mouse sperm undergoes hyperpolarization during capacitation. Biol Reprod. 2015 May;92(5):121. doi: 10.1095/biolreprod.114.127266
Abi Nahed R et al. Slo3 K+ channel blocker clofilium extends bull and mouse sperm-fertilizing competence. Reproduction. 2018 Dec 1;156(6):463-476. doi: 10.1530/REP-18-0075 (This article could be of use to enhance one of the possible applications of the study)
Response:We are thankful for your comments in enriching the discussion of the manuscript.
Line 251-253: “Additionally, the crucial role of Slo3 in human sperm physiology is highlighted when the function of the sperm is impaired by Slo3 specific channel blocker [22].”
Line 265-268: “In agreement with previous studies on human [4] and mouse [35] sperm, only a subpopu-lation of boar sperm respond to membrane potential changes. Hence, the subpopulation responding via membrane potential was initially identified and then gated using a Valinomycin-KCl setup.”
Line 416-420: “This study provides a significant contribution to the literature as identification and mechanistic underpinning of boar sperm KSper is paramount in improving the success of fertilization and sperm handling at animal farms as evident from other animal studies (i.e., bull) [53] wherein specific blockers targeting Slo3 could extend the fertilizing competence.”
MINOR COMMENTS
- Line 46 Please replace “Capacitated sperm” with “In capacitated sperm”
Line 55, please consider replacing “homologous” with homology
Response:Thank you very much for your comment and carefully reviewing the manuscript. We have made the necessary edits in line 41 and 51 of the manuscript including the suggested replacements.

Reviewer 3 Report
Overall, the study was well planned and carried through; the information given in the text provides a good background as well as perspectives of the Slo3 is the principal channel participating in membrane potential modulations and it is also involved in sperm motility, intracellular ion homeostasis, and acrosome reaction.
Comments and Suggestions for Authors
Line 97: The authors mention the Total motility of sperm increases with LDD175 up to nearly 80% when incubated for 5mins. However, the results are not shown in Figure 1.
Author Response
Reviewer #3
We appreciate your time and effort put into reviewing and improving the quality of our manuscript. We revised our manuscript incorporating your thoughtful comments and suggestions. Followings are our point-by-point responses to each of your comments.
- Line 97: The authors mention the Total motility of sperm increases with LDD175 up to nearly 80% when incubated for 5mins. However, the results are not shown in Figure 1.
Response: Thank you very much for your comment and for carefully reviewing the manuscript. We have changed Figure 1C to be more visible for the motile and immotile populations. The detailed information is in the results section for figure 1.
